# Approval Research for Carcinogen Humic-Like Substances (HULIS) Emitted from Residential Coal Combustion in High Lung Cancer Incidence Areas of China

**Kai Xiao [1], Qingyue Wang [1,\*], Yichun Lin [1], Weiqian Wang [1], Senlin Lu [2,\*] and Shinich Yonemochi [3]**

[1] Graduate School of Science and Engineering, Saitama University, 255 Shimo-Okubo, Sakura-ku, Saitama 338-8570, Japan; xiao.k.662@ms.saitama-u.ac.jp (K.X.); lin.y.852@ms.saitama-u.ac.jp (Y.L.); weiqian@mail.saitama-u.ac.jp (W.W.)

[2] School of Environmental and Chemical Engineering, Shanghai University, 99 Shangdalu, Baoshan District, Shanghai 200444, China

[3] Center for Environmental Science in Saitama, 914 Kamitanadare, Kazo 347-0115, Japan; yonemochi.shinichi@pref.saitama.lg.jp

\* Correspondence: seiyo@mail.saitama-u.ac.jp (Q.W.); senlinlv@staff.shu.edu.cn (S.L.)

**Abstract:** The incidence and mortality rate of lung cancer is the highest in Xuanwei County, Yunnan Province, China. The mechanisms of the high lung incidence remain unclear, necessitating further study. However, the particle size distribution characteristics of HULIS emitted from residential coal combustion (RCC) have not been studied in Xuanwei. In this study, six kinds of residential coal were collected. Size-resolved particles emitted from the coal were sampled by using a burning system, which was simulated according to RCC made in our laboratory. Organic carbon (OC), elemental carbon (EC), water-soluble inorganic ion, water-soluble potentially toxic metals (WSPTMs), water-soluble organic carbon (WSOC), and HULIS-C (referred to as HULIS containing carbon contents) in the different size-segregated particulate matter (PM) samples were determined for health risk assessments by inhalation of PM. In our study, the ratio of HULIS-Cx to WSOCx values in RCC particles were 32.73–63.76% (average 53.85 $\pm$ 12.12%) for $PM_{2.0}$ and 33.91–82.67% (average 57.06 $\pm$ 17.32%) for $PM_{2.0\sim7.0}$, respectively. The carcinogenic risks of WSPTMs for both children and adults exceeded the acceptable level ($1 \times 10^{-6}$, indicating that we should pay more attention to these WSPTMs). Exploring the HULIS content and particle size distribution of the particulate matter produced by household coal combustion provides a new perspective and evidence for revealing the high incidence of lung cancer in Xuanwei, China.

**Keywords:** residential coal combustion; HULIS-C; WSOC; carcinogenic risks; Xuanwei

## 1. Introduction

Atmospheric particulate matter has received more attention as air pollution has caused about seven million premature deaths every year. Fine particles ($PM_{2.0}$ or smaller) are generally more dangerous and ultrafine particles (one micron in diameter or less), which can penetrate tissues and organs, pose a greater risk of stroke, heart disease, chronic obstructive pulmonary disease, lung cancer and acute respiratory (https://www.who.int/; accessed on 18 May 2021). Solid fuels are still the main energy source for cooking and heating in some rural areas of China. In 2019, about 2.7 billion tons of coal were consumed in China, which contributed about 68.6% of the national primary energy source (http://www.stats.gov.cn/tjsj/ndsj/; accessed on 18 May 2021). RCC are important anthropogenic sources of particulate and toxic and hazardous pollutants in the atmosphere [1]. The particles generated from RCC produce reactive oxygen species (ROS), especially once it enters the human body. Oxidative stress in the airways and alveoli leads to stimulation of alveolar macrophages and injury to the epithelial lining, which in turn attracts inflammatory cells from the circulation [2]. This mechanism is considered to be related to highly redox-active

components in the particulate matter. The identified amounts of metals and quinones are responsible for only a very small mass concentration of particulate matter in the atmosphere. Recently, an abundant water-soluble organic component, namely HULIS, has been identified as highly redox-active and contributes to the oxidation potential of particulate matter [3,4].

Humic-like substances are large molecules of unresolved water-soluble organic carbon with a polycyclic ring or carboxyl, carbonyl, and hydroxyl group structure [5], consisting of 9–72% of WSOC [6]. They have been widely reported in atmospheric aerosols in urban [7], rural, forested [8], and marine environments [9], as well as rainwater [10], cloud water [11], and fog [12], with concentrations varying over two orders of magnitude from below 0.1 μg m$^{-3}$ to dozens of μg m$^{-3}$ [7]. HULIS was found to enhance catalytically the generation of ROS under simulated physiological conditions, thereby likely may contribute to PM-related health problems [13–15]. Many studies have shown that biomass combustion and secondary formation in the atmosphere are considered important sources of HULIS [8,16–18]. As for their health effects, redox sites on HULIS can promote electron transfer, leading to the production and conversion of excess ROS during respiratory deposition, thus disrupting the redox homeostasis of affected cells [3,17]. Subsequent intracellular stimulation of oxidative stress may activate downstream pathways leading to various respiratory and cardiovascular diseases [19]. It has recently been suggested that RCC is a significant and major source of HULIS [5], which has been accepted as the main cause of human lung cancer [20–23].

Xuanwei has the highest incidence and mortality rate of lung cancer in China [1]. According to the "2012 Chinese Cancer Registry Annual Report," the average incidence rate of lung cancer in Xuanwei was 92 per 100,000, which is four–five times higher than the national rate [24]. The incidence of lung cancer among non-smokers is 400/100,000 [25,26], which is 20 times higher than the national average, especially among women in rural areas [20,27,28]. However, the particle size distribution characteristics of HULIS mitted from RCC has not been studied in Xuanwei.

Rural houses in the Xuanwei area are mainly two-story civil structures (Figure 1a). The first half of the first floor of this type of house is the living room or kitchen, which is the place where the family often stays. Due to the small windows installed, the air circulation is not smooth. As a result, smoke is generated when residents use local coal for heating or cooking. As shown in Figure 1b, the stove is directly discharged indoors without any treatment, so the concentration of particulate matter is very high. The link between the high incidence of lung cancer and harmful pollutants emitted from local solid fuel combustion has been a hot topic and focus of research since the 1980s in Xuanwei, Yunnan Province, China [25]. However, the mechanisms of the high lung incidence remain unclear, necessitating further study.

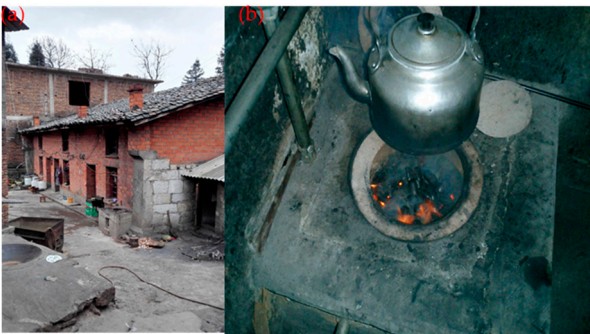

**Figure 1.** The stove (**a**) and building structure (**b**) in rural Xuanwei, China.

The information about HULIS has not been reported within high-incidence areas of lung cancer in Xuanwei. In this study, we selected six types of coal collected from six kinds of mines in Xuanwei and conducted simulated combustion experiments to explore the

content and particle size distribution pattern of HULIS-C and health risk assessment of WSPTMs in particulate matter produced by RCC, providing new perspectives and evidence to reveal the high prevalence of lung cancer in Xuanwei.

## 2. Materials and Methods

After field investigation, we collected raw coal from three areas, including six kinds of residential coal (Table 1) from Zhangyi District (Luomu coal (LM) and Bole coal (BL)), Fuyuan County (Houshou Town, Lijiawu coal (LJW); Laochang Town, Shunfa coal (SF)), and Laibin Town in Xuanwei City (Guangming coal (GM) and Zongfan coal (ZF)). The information about the sampling site is shown in Figure 2. All collected samples were stored in plastic bags at the sampling site to avoid contamination or oxidation of the samples. The residential coals were collected according to the methods [29].

**Table 1.** Residential coal samples information.

| Sites | Sample | Mine | Altitude/m | Latitude | Longitude |
|---|---|---|---|---|---|
| Zhanyi | BL | BL mine | 2104 | 25°47′15.32″ | 104°07′32.32″ |
| | LM | LM mine | 1793 | 26°29′34.09″ | 103°46′9.17″ |
| Fuyuan | SF | SF mine | 1994 | 25°13′31.13″ | 104°31′22.42″ |
| | LJW | LJW mine | 2078 | 25°79′99.21″ | 104°28′60.06″ |
| Xuanwei | GM | GM mine | 1987 | 26°19′46.55″ | 104°09′36.43″ |
| | ZF | ZF mine | 2024 | 26°17′58.25″ | 104°05′42.49″ |

(Bole coal (BL), Luomu coal (LM), Shunfa coal (SF), Lijiawu coal (LJW), Guangming coal (GM), and Zongfan coal (ZF)).

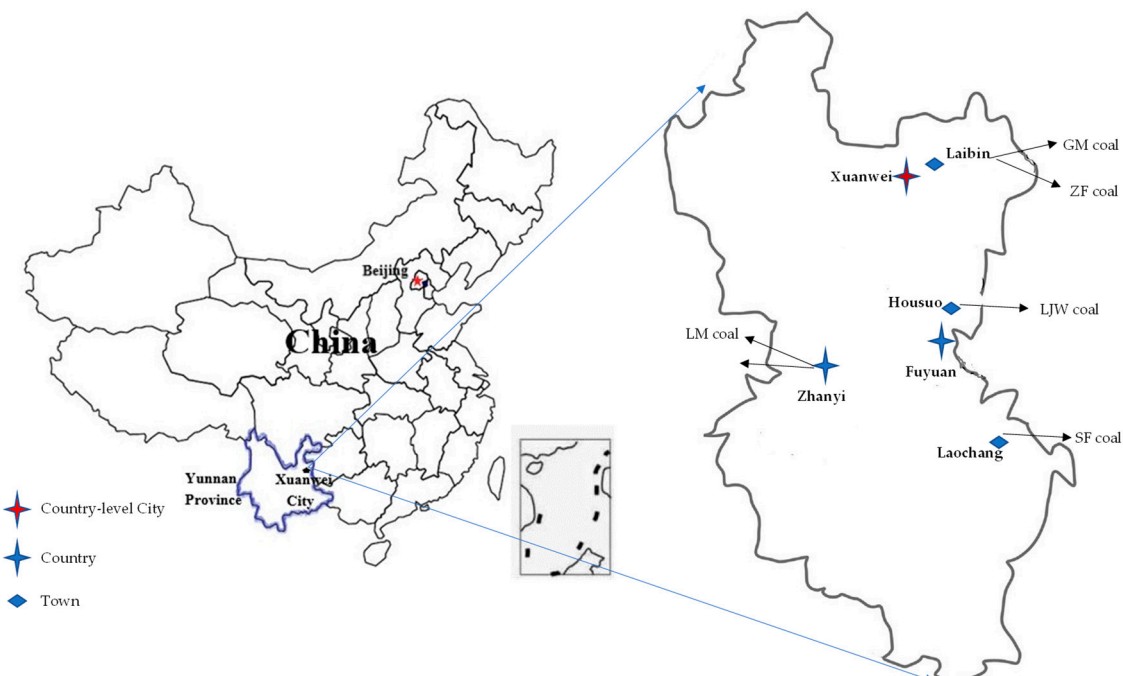

**Figure 2.** Sampling sites (Bole coal (BL), Luomu coal (LM), Shunfa coal (SF), Lijiawu coal (LJW), Guangming coal (GM), and Zongfan coal (ZF)).

### 2.1. Sample Processing and Collection of Simulated Combustion Particulate Matter

The flue gas collection system consists of a flue device and sampler. The flue device consists of a closed trumpet-shaped hood, two elbows, and a cooling water sleeve. In addition, a closed flue gas holding chamber is connected to the end of the flue, and Andersen high volume air sampler (Shibata Science Co., Ltd., Saitama., Japan) is placed in it. Several parts of the sampling system can be disassembled for easy cleaning, and they

are connected by flanges. In order to avoid the pollution of the flue material to the flue gas, the entire flue device is made of stainless steel. The gasket used in the connection part is Teflon material (Figure 3).

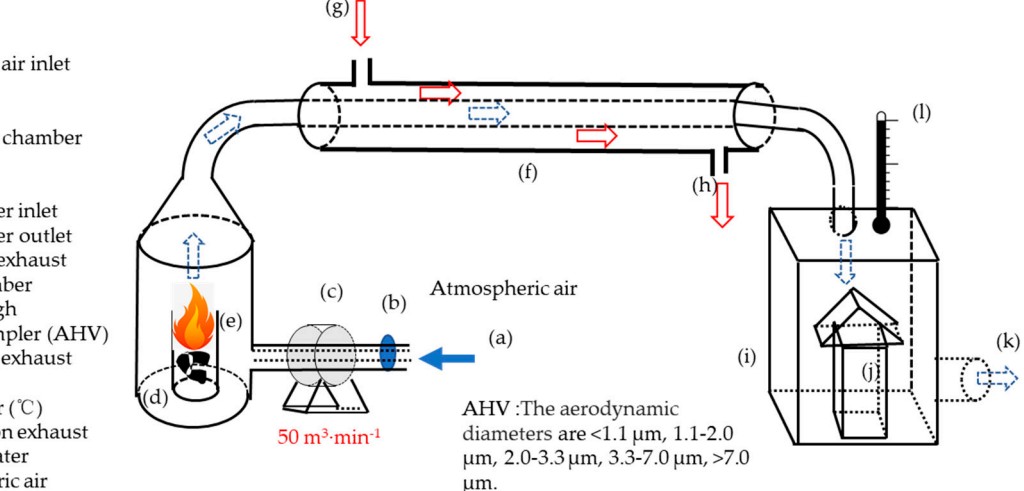

**Figure 3.** Sketch of sampling system.

Before simulating combustion, all parts of the sampling system are carefully cleaned. The solid fuel block is ignited with alcohol in the furnace and the coal furnace is placed directly under the horn hood. The closed horn-shaped hood introduces clean air through the blower. Minimizing the background pollution during sampling is convenient for experimental analysis. The flue gas is fully mixed by the flue device through the function of the sampler's pump and blower, and the temperature is reduced at the cooling water jacket. Finally, the flue gas temperature at the sampler is controlled at about 40 °C. During the experiment, it is important to ensure that there is no flue gas leakage during the entire coal combustion process and that there is no obvious particle deposition in the pipeline.

Pre-treatment of filters has been consistent in our previous studies [30]. Before and after each sampling, the filters were weighed by an electronic balance with a detection limit of 0.01 mg (Sartorius, Co., Ltd., Göttingen., Germany). The Andersen sampler was placed horizontally in the flue gas holding chamber. The particles in five sizes were collected, with a flow rate at 566 L/min, and the aerodynamic diameters were <1.1 μm, 1.1–2.0 μm, 2.0–3.3 μm, 3.3–7.0 μm, and >7.0 μm, respectively. In this study, the Anderson sampler used did not have cut-off sizes of 2.5 and 10 μm, and fine-mode and coarse-mode particles were defined as particles smaller than 2.0 and >2.0 μm, respectively. After sampling, the filter membrane was wrapped with the original aluminum foil paper, the sample of the filter membrane was weighed after constant temperature and humidity for 24 h, then wrapped with aluminum foil and stored frozen until further analysis.

### 2.2. Analytical Method of Water-Soluble Ionic Species and Water-Soluble Potentially Toxic Metals

The method has been described in our previous study [30]. Briefly, normally the extraction rate of sulfate, nitrate, and ammonium can reach more than 98%. After weighing, portions of the sample filters (including three blank filters) submerged in a vial were ultrasonically extracted with 10 mL ultrapure water (resistivity > 18 MΩcm$^{-1}$) and repeated three times, each time for 10 min. The aqueous extract solution was filtered with a pore size of 0.20 μm polytetrafluoroethylene (PTFE; DISMIC-13$_{HP}$) membranes and then the concentrations of four anions (Chlorine ion, $Cl^-$; Nitrate ion, $NO_3^-$; Nitrite ion, $NO_2^-$; Sulfate ion, $SO_4^{2-}$) and five cations (Ammonium ion, $NH_4^+$; Sodium ion, $Na^+$; Potassium ion, $K^+$; Calcium ion, $Ca^{2+}$; and Magnesium ion, $Mg^{2+}$) were analyzed by ion chromatography (IC) [31]. Meanwhile, the solution after filtration, 10 WSPTMs (V, Cr, Mn, Co, Ni, Zn, As, Cd, Ba, and Pb), was analyzed by inductively coupled plasma mass spectrometer (ICP-MS)

to determine the concentrations at the Center for Environmental Science in Saitama (CESS) in Japan ($\mu$g m$^{-3}$).

### 2.3. Analytical Method of Organic Carbon (OC) and Elemental Carbon (EC)

Total organic carbon (TOC) analyzer (multi-N/C 3100 TOC Analytik Jena, Jena., Germany) was used to analyze the OC and EC with the IMPROVE_A heating procedure (Table 2) [31,32]. The sample was heated in a non-oxidizing helium (He) atmosphere at 120 °C ($OC_1$), 250 °C ($OC_2$), 450 °C ($OC_3$), and 550 °C ($OC_4$), and in an oxidizing atmosphere with 2% oxygen in helium equilibrium by gradually heating an aliquot of the punch of the sample quartz filter at 550 °C ($EC_1$) and 700 °C ($EC_2$). OC can evolve as pyrolysis carbon (OP) during the heating process. The IMPROVE-A thermal/optical protocol defines that $OC = OC_1 + OC_2 + OC_3 + OC_4 + OP$; $EC = EC_1 + EC_2 + EC_3 - OP$; $TC = OC + EC$ [33]. The detection limits of OC and EC were below 1.0 $\mu$g m$^{-3}$, which has been described in detail in a pervious study [34]. Replicate analyses were performed once every 10 samples. The determined repeatability was better than 5% [35].

**Table 2.** Heating process and temperature in each stage of OC and EC.

| Fration | Pyrolized Fration | Temperature Range (°C) | Atmosphere |
|---|---|---|---|
| $OC_1$ | | Ambient to 120 | |
| $OC_2$ | | 120–150 | |
| $OC_3$ | | 150–250 | 100% He |
| $OC_4$ | | 250–450 | |
| OPC | | 450–550 | |
| $EC_1$ | OP | Remains at 550 | |
| $EC_2$ | | 550–700 | 98% He, 2% $O_2$ |
| $EC_3$ | | 700–800 | |

### 2.4. Isolation and Measurement of Water-Soluble HULIS-C and WSOC

Many published studies have shown that HULIS is usually isolated by solid phase extraction (SPE) [31,36–38]. The flow chart of HULIS-C (black arow) and WSOC (blue arow) isolation and measurement are illustrated in Figure 4. To obtain the water-soluble fraction of PMs (a,b), as described in Section 2.3, the water extracts were filtered through a syringe PTFE filter with a 0.20 $\mu$m PTFE (Figure 4c) to avoid any interference (insoluble suspensions, filter debris) from residual quartz filters [13]. Then, 10 mL of the aqueous extract was acidified to pH = 2 with 1 M hydrochloric acid and loaded onto a pretreated C-18 solid phase extraction column (octadecyl carbon chain bonded silica gel; 200 mg, 3 mL) to separate into HULIS and hydrophilic fractions (Figure 4d). Subsequently, C-18 was rinsed with two portions of 1 mL ultrapure water before elution with 4.5 mL of high-performance liquid chromatography grade methanol containing 2% ammonia (w/w) (Figure 4e). Finally, the eluate was dried under a gentle nitrogen stream for 5–7 h (Figure 4f) and re-redissolved in 10 mL of ultrapure water (Figure 4g) and then analyzed by a total organic carbon analyzer (Multi N/C 3100 TOC, Analytik Jena, Jena., Germany). (Figure 4h). The details of WSOC were described in a previous study (Figure 4a–c,h) [39,40]. WSOC were measured by a total organic carbon analyzer. In this study, HULIS-C refers to HULIS with carbon content, and the OC/Organic Matter (OM) conversion factor was not used to convert the quality of HULIS due to the uncertainty of the OC/OM conversion factor method and the lack of OC/OM conversion factors for RCC particles. Therefore, the generic term "HULIS" is used to interpret some results and discussions.

### 2.5. Assessment Method for Human Health Risk

This study evaluated the risk to human health of WSPTMs to better understand the RCC particles via direct inhalation to residents. According to the United States Environmental Protection Agency (US EPA), Integrated Risk Information System (IRIS), and the International Agency for Research on Cancer (IARC), arsenic, cadmium, cobalt, chromium,

cadmium, nickel, and lead are human carcinogens. Arsenic, barium, cadmium, cobalt, chromium, nickel, manganese, and vanadium (VI) are reported as having non-carcinogenic hazard effects [41].

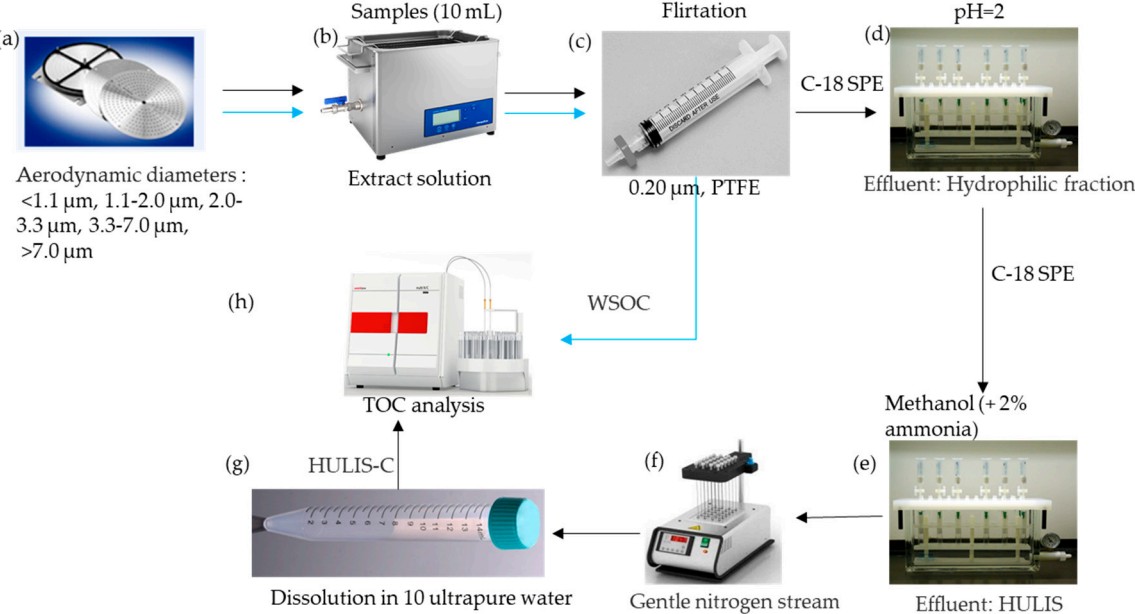

**Figure 4.** The flow chart of HULIS-C (black arow) and WSOC (blue arow) isolation and measurement. ((**a**): Cutting saples; (**b**): Extract solution; (**c**): Flirtation; (**d**): Separate HULIS and hydrophilic fractions; (**e**): Separate HULIS and hydrophilic fractions; (**f**): Isolation HULIS; (**g**): Redissolved HULIS; (**h**): HULIS and WSOC analysis).

The methodology has been used in previous studies [30,42]. Sensitive local residents were divided into two groups (i.e., children and adults). Thus, inhalation exposure concentration (*IEC*), hazard quotient (*HQ*) for non-carcinogenic risk, and carcinogenic risks (*CR*) of WSPTMs in RCC were calculated following Equations (1)–(6):

$$IECn = \frac{Ci \times ET \times EF \times ED}{ATn} \tag{1}$$

$$IECc = \frac{Ci \times ET \times EF \times ED}{ATc} \tag{2}$$

$$HQi = \frac{EC_n}{RfC \times 1000 \ \mu g \ mg^{-1}} \tag{3}$$

(i = As, Cd, Co, Cr (VI), Mn, Ni, V, Zn, Ba and, Ba)

$$CRi = IUR \times IECc \ 4) \tag{4}$$

(i = V, As, Cd, Cr (VI), Ni, Co, and Pb)

$$TCR = \Sigma CRi \tag{5}$$

$$HI = \Sigma HQi \tag{6}$$

The inhalation exposure concentration (*EC*) is each element examined through inhalation ($\mu g \ m^{-3}$), where Ci is the average concentration of individual WSPTMs of PM size i, ECn is used to calculated the values of *HQ*, ECc is used to calculated the values of *CR*, *TCR* is the total carcinogenic risks of all the WSPTMs, *HI* is the total hazard quotient of all the WSPTMs, and *AT*, *ED*, *EF*, and *ET* are average lifetime (hours), exposure duration (year), exposure frequency (days year$^{-1}$), and exposure time (h day$^{-1}$), respectively. The values of

all the parameters and explanations are listed in Table 3. The parameter of inhalation unit risk (*IUR*) and inhalation reference dose (*RfC*) value of potentially toxic metals are shown in Table 4. The *CR* value between $1 \times 10^{-6}$ and $1 \times 10^{-4}$ indicates acceptable or tolerable carcinogenic risk. If the value is higher than $11 \times 10^{-4}$, it means the risk is unacceptable. A *CR* value lower than $1 \times 10^{-6}$ indicates significant health hazards.

**Table 3.** Exposure parameters [43] of health risk assessment.

| Exposure Parameter | Value for Children | Value for Adults |
|---|---|---|
| *EF* (exposure frequency) | 350 day·year$^{-1}$ | 350 day·year$^{-1}$ |
| *ET* (exposure time) | 24 h·day$^{-1}$ | 24 h·day$^{-1}$ |
| *ED* (exposure duration) | 6 years | 24 years |
| *ATn* (average time for non-carcinogenic) | ED × 365 days·year$^{-1}$ × 24 h·day$^{-1}$ | ED × 365 days·year$^{-1}$ × 24 h·day$^{-1}$ |
| *ATc* (average time for carcinogenic) | 74.83 year × 365 days· year$^{-1}$ × 24 h·day$^{-1}$ | 74.83 year × 365 days· year$^{-1}$ × 24 h·day$^{-1}$ |

China's Sixth National Census Shows Average Life Expectancy Reaches 74.83 Years. (National Bureau of Statistics of the People's Republic of China; http://www.stats.gov.cn/tjsj/ndsj/; accessed on 13 June 2021).

**Table 4.** *RfC* and *IUR* values [44] for different water-soluble potentially toxic metals (WSPTMs).

| WSPTMs | *IUR* (µg m$^{-3}$)$^{-1}$ | *RfC* (mg m$^{-3}$) |
|---|---|---|
| V | $8.3 \times 10^{-3}$ | $1.0 \times 10^{-4}$ |
| Cr (VI) | $1.2 \times 10^{-2}$ | $1.0 \times 10^{-4}$ |
| Co | $9.0 \times 10^{-3}$ | $6.0 \times 10^{-6}$ |
| Ni | $2.4 \times 10^{-4}$ | $1.4 \times 10^{-5}$ |
| As | $4.3 \times 10^{-3}$ | $1.5 \times 10^{-5}$ |
| Cd | $1.8 \times 10^{-3}$ | $1.0 \times 10^{-5}$ |
| Pb | $1.2 \times 10^{-5}$ | - |
| Zn | - | $3.01 \times 10^{-1}$ |
| Ba | - | $5.00 \times 10^{-4}$ |
| Mn | - | $5.00 \times 10^{-5}$ |

*2.6. Statistics*

Origin 8.0 (OriginLab Corporation., Northampton, MA, USA) was used to draw the figures in this article.

## 3. Results and Discussion

Mineral particles have been identified as hazardous to the respiratory system, and RCC particles are a major contributor in Xuanwei [21,45,46].

*3.1. Mass Concentrations of RCC Particles*

Based on the simulated six kinds of RCC data, the mass percentage mean value of $PM_{1.1}$, $PM_{1.1-2.0}$, $PM_{2.0-3.3}$, $PM_{3.3-7.0}$, and $PM_{>7.0}$ were 38.67 ± 6.19%, 28.35 ± 4.34%, 15.87 ± 6.18%, 9.94 ± 3.51%, and 7.16 ± 3.51%, respectively. As shown in Figure 5, the mass percentages for $PM_{1.1}$, $PM_{1.1-2.0}$, $PM_{2.0-3.3}$, $PM_{3.3-7.0}$, and $PM_{>7.0}$ ranged from 29.21 ± 49.48%, 22.75 ± 34.66%, 10.04 ± 28.61%, 6.21 ±16.42%, and 4.09 ± 14.41%, respectively. The particulate matters emitted from RCC were mainly distributed in the size range of <2.1 µm, which accounted for 52.45–77.16% of total particulate matter. Compared to the previous results [47], the particulate matters emitted from RCC (Huaibei, Xinjiang, Inner Mongolia, and Guizhou in china) were mainly concentrated in the size range of 0.43–2.1 µm, which accounted for 39.2–62.8% of total particulate matter, indicating that the fine particulate matters emitted from RCC in the Xuanwei area are higher than other areas of China.

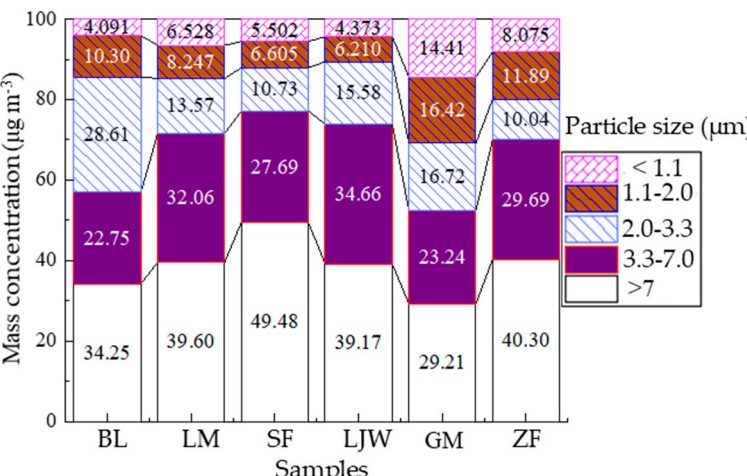

**Figure 5.** Mass concentration percentages of particles emitted from RCC (%) (Bole coal (BL), Luomu coal (LM), Shunfa coal (SF), Lijiawu coal (LJW), Guangming coal (GM), and Zongfan coal (ZF)).

### 3.2. Size Distribution of Ion Species

Sources and size distribution characteristics of ions species are better known and can serve as valuable references for the lesser-known aerosol constituents.

Figure 6 shows the mass concentrations ($\mu g \, m^{-3}$) and standard deviation (SD) of ion species (coal, N = 6). As mentioned in Section 3.1, five size-segregated PM filters in each RCC samples were analyzed for mass concentrations and therefore we showed the mass concentrations of ion species with five various size ranges. As can be seen from Figure 6, coal combustion emits more $NH_4^+$ and $SO_4^{2-}$ [48]; the average concentrations of individual ions were in the order of $SO_4^{2-}$ (47.46 $\mu g \, m^{-3}$) > $Cl^-$ (20.61 $\mu g \, m^{-3}$) > $NO_3^-$ (10.14 $\mu g \, m^{-3}$) > $NH_4^+$ (6.77 $\mu g \, m^{-3}$) > $NO_2^-$ (47.46 $\mu g \, m^{-3}$) > $Na^+$ (5.07 $\mu g \, m^{-3}$) > $Mg^{2+}$ (2.32 $\mu g \, m^{-3}$) > $K^+$ (1.93 $\mu g \, m^{-3}$) > $Ca^{2+}$(0.96 $\mu g \, m^{-3}$), respectively (Table S1). The size distribution of nine water-soluble inorganic ions (WSIIs) were mainly concentrated in fine particles. Secondary inorganic aerosols (SIA, $SO_4^{2-}$, $NO_3^-$, and $NH_4^+$), the major components of water-soluble inorganic ions (WSIIs), contributed to 59.08 ± 9.48% of total WSIIs (Table S1). Compared with previous studies, WSIIs/$PM_{2.1}$ is 2.22 ± 2.36% lower than atmospheric particulate matter occupied by 33–41% or more of fine particles [47,48]. Sulfate dominated SIA was extremely high and coal combustion was a candidate of high sulfate concentrations [49,50], which is consistent with our research (as Figure 6a). On the other hand, the average charge of the anion in each particle size segment is 2.72 ± 1.17 times more than the cation (Table S2), indicating that the aqueous solution of the particulate matter emitted by the residential coal burning is acidic in the Xuanwei area.

This experiment was carried out in a laboratory. As we all know, there is a big difference between laboratory experiments and field measurements. In this study, there is no obvious relationship between temperature and humidity with WSIIs. Air supply and burning condition were the parameters influencing the emissions of pollutants from coal combustion [51]. $NH_4^+$ closely correlated with $SO_4^{2-}$ was provided in the supplementary material (Figure S1), which may indicate the complete neutralization of $SO_4^{2-}$ by $NH_4^+$ in particulate matter. It can be inferred that the combustion of residential coal can result in the formation of considerable particulate sulfate compounds [52], which can change the radiation balance by scattering or absorbing solar radiation and thermal radiation from the earth's surface [53].

### 3.3. The Characteristics of EC, OC, WSOC, and HULIS-C in RCC Particles

WSOC makes up a significant portion of OC, accounting for 10–90% depending on location, source, and meteorological conditions [54]. HULIS-C are important hydrophobic compounds of WSOC in ambient aerosols.

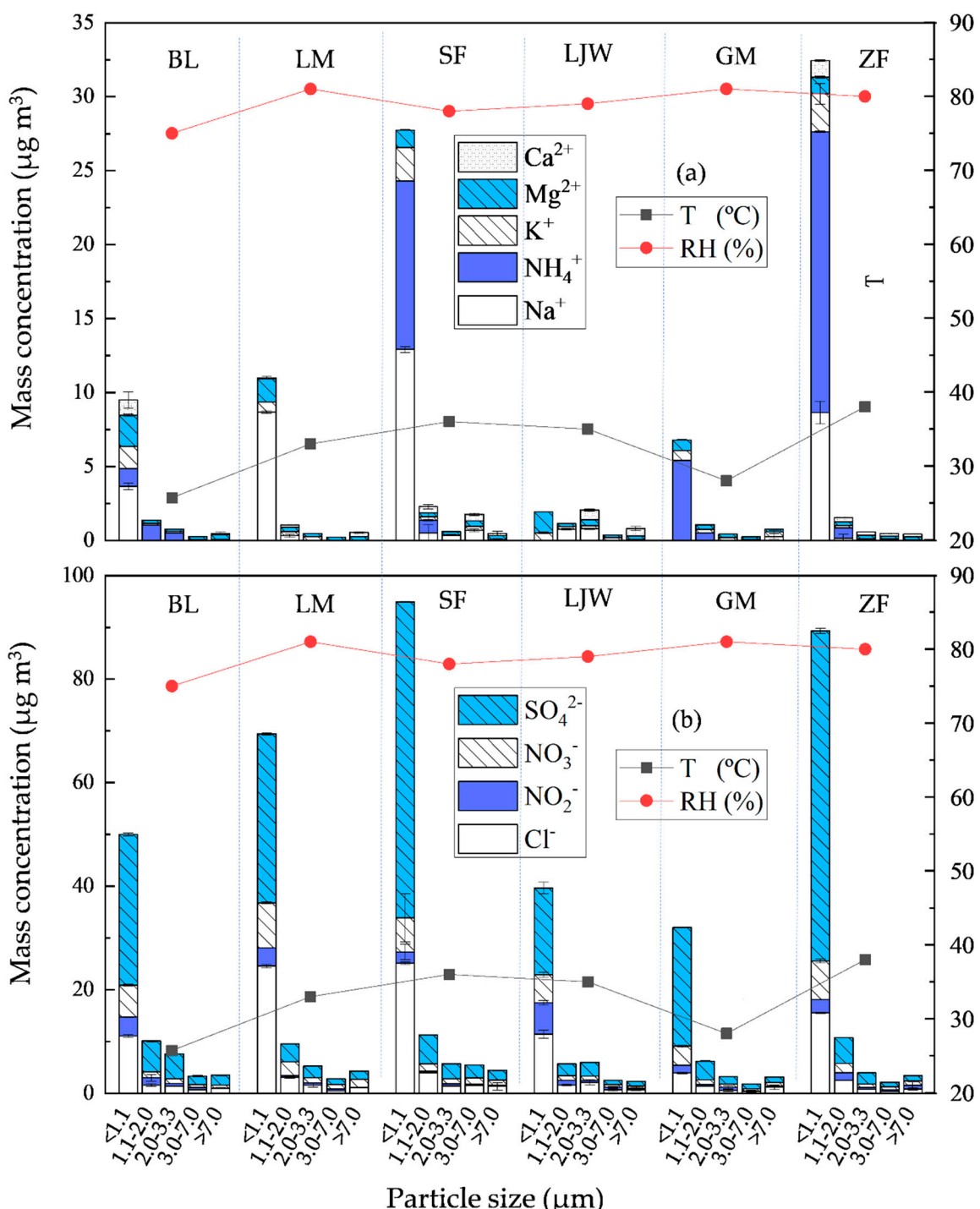

**Figure 6.** Mass concentration ($\mu g\ m^{-3}$) and standard deviation (STD) of ion species (Coal, N = 6, (**a**): cation mass concentration; (**b**): anion mass concentrations) (Bole coal (BL), Luomu coal (LM), Shunfa coal (SF), Lijiawu coal (LJW), Guangming coal (GM), and Zongfan coal (ZF)).

### 3.3.1. Abundance of EC, OC, WSOC, and HULIS-C in RCC Particles

Research based on the Community Multiscale Air Quality (CMAQ) model (version 5.0.1) shows that RCC is an important source of HULIS, accounting for 15.1% in Beijing [55]. Mass concentrations of OCx, ECx, WSOCx, and HULIS-C in coarse and fine PM from RCC particles are shown in Table 5. The average values of WSOC, HULIS-C, OC, and EC were $234.72 \pm 149.04\ \mu g \cdot m^{-3}$, $117.65 \pm 53.90\ \mu g\ cm^{-3}$, $3250.99 \pm 2126.65\ \mu g \cdot m^{-3}$, and

$643.19 \pm 263.94$ µg·m$^{-3}$ in PM$_{2.0}$, while they were $82.37 \pm 46.82$ µg·m$^{-3}$, $40.23 \pm 19.17$ µg cm$^{-3}$, $822.86 \pm 522.48$ µg·m$^{-3}$, and $30.11 \pm 35.62$ µg·m$^{-3}$ in PM$_{2.0\sim7.0}$.

**Table 5.** Concentrations of OCx, WSOCx, and HULIS-C in PM$_{2.0}$ and PM$_{2.0-7.0}$ from RCC particles.

| Samples | Particle Size µm | OC | EC | TC | PM | HULIS-C | WSOC |
|---|---|---|---|---|---|---|---|
| | | µg m$^{-3}$ | | | | | |
| BL | | 1556.00 | 749.35 | 2305.35 | 3651.35 | 137.78 | 206.40 |
| LM | | 2385.33 | 851.68 | 3237.01 | 7854.83 | 168.63 | 275.95 |
| SF | PM$_{2.0}$ | 4490.54 | 353.95 | 4844.49 | 10,303.30 | 137.19 | 215.44 |
| LJW | | 6983.05 | 1056.47 | 8039.52 | 11,833.04 | 174.01 | 531.67 |
| GM | | 3649.02 | 503.15 | 4152.17 | 4185.81 | 38.76 | 90.00 |
| ZF | | 442.01 | 344.56 | 786.57 | 1810.95 | 49.52 | 88.85 |
| Average | | 3250.99 | 643.19 | 3894.18 | 6606.55 | 117.65 | 234.72 |
| TSD | | 2126.65 | 263.94 | 2264.47 | 3654.48 | 53.90 | 149.04 |
| BL | | 1000.09 | 108.73 | 1108.82 | 2492.64 | 71.85 | 108.53 |
| LM | | 1023.89 | 18.13 | 1042.02 | 2391.05 | 44.02 | 72.80 |
| SF | PM$_{2.0-7.0}$ | 959.05 | 22.78 | 981.82 | 2314.49 | 45.92 | 135.41 |
| LJW | | 1632.95 | 16.60 | 1649.55 | 3492.34 | 46.40 | 131.92 |
| GM | | 244.74 | 6.04 | 250.78 | 1311.84 | 17.84 | 21.69 |
| ZF | | 76.47 | 8.35 | 84.82 | 1143.99 | 15.35 | 23.86 |
| Average | | 822.86 | 30.11 | 852.97 | 2191.06 | 40.23 | 82.37 |
| TSD | | 522.48 | 35.62 | 533.08 | 786.23 | 19.17 | 46.82 |

(Bole coal (BL), Luomu coal (LM), Shunfa coal (SF), Lijiawu coal (LJW), Guangming coal (GM), and Zongfan coal (ZF)).

There is a similar contribution of HULIS-C to the PM, which was 2.09–5.65% for PM$_{2.0}$ and 2.68–5.62% for PM$_{2.0-7.0}$, respectively. This result was consistent with previous research where HULIS accounted for $3.5 \pm 0.4\%$ of PM$_{2.5}$ emitted from coal combustion [36] but was lower than the reported WSOC results (14–56%) for biomass burning PM$_{2.5}$, which was significantly lower than that ($22.6 \pm 3.7\%$) of ambient PM$_{2.5}$ [36].

Figure 7 show the mass concentration (µg m$^{-3}$) and standard deviation (STD) of OC, EC, WSOC, and HULIS-C (Coal, N = 6) in RCC particles. The mass concentrations of HULIS are in the following order: 1.1 µm > 1.1–2.0 µm > 2.0–3.3 µm > 3.3–7.0 µm. The mass concentration of HULIS-C in our research far exceeded HULIS-C in the ambient average range of 0.8–15.9 µg·m$^{-3}$ [8,15,31,55–57]. The variability in mass concentration can be explained by the independent and closed sampling system. It is also related to the type and maturity of the coal. Relatively little water-soluble brown carbon was found in particles produced by burning middle-maturity coal in residential stoves, while significant amounts were found in low maturity bituminous coal and anthracite [58]. Unlike general atmospheric particulate sampling, we collected high concentrations of RCC particulate matter directly with our sampling system. In the future, a comprehensive investigation of coal combustion HULIS-C emissions under different stove types, combustion conditions, and combustion stages is necessary to better understand the distribution pattern of HULIS-C.

### 3.3.2. Size Distribution of HULIS-C in RCC Particles

HULIS-C accounts for a large proportion of WSOC in RCC particles and has received much attention in recent years [58]. The percentage of OCx, ECx, WSOCx to WSOC, and HULIS-C in coarse and fine PM from RCC particles is show in Table 6. In our study, the HULIS-Cx/WSOCx (%) values in RCC particles were 32.73–63.76% (average $53.85 \pm 12.12\%$) for PM$_{2.0}$ and 33.91–82.67% (average $57.06 \pm 17.32\%$) for PM$_{2.0-7.0}$, respectively. This suggests that HULIS-C is the main component of WSOC in RCC particles. Despite the large variance of HULIS-C concentrations, the abundance of HULIS-C in WSOC was relatively stable. Interestingly, the HULIS-Cx/HULIS-Ct (%) values in RCC particles were 68.48–79.30% (average $73.95 \pm 5.13\%$) for PM$_{2.0}$ and 20.70–34.27% (average $26.05 \pm 5.13\%$) for PM$_{2.0-7.0}$, respectively. The first study about HULIS percentage contributions in particles in the Pearl River Delta region in China reported similar

results, where HULIS had the highest percentage contribution (81%) in $PM_{0.63-0.87}$ and the lowest percentage contribution (7%) in $PM_{4.0-5.7}$ during biomass burning seasons [59]. Our conclusion indicates that the HULIS-C emitted from RCC is mainly concentrated in fine particles, and the mass concentration of HULIS-C and WSOC is inversely proportional to the particle size (Pearson correlation coefficients $0.92 \pm 0.16$) in our research (Figure 7, Table S3).

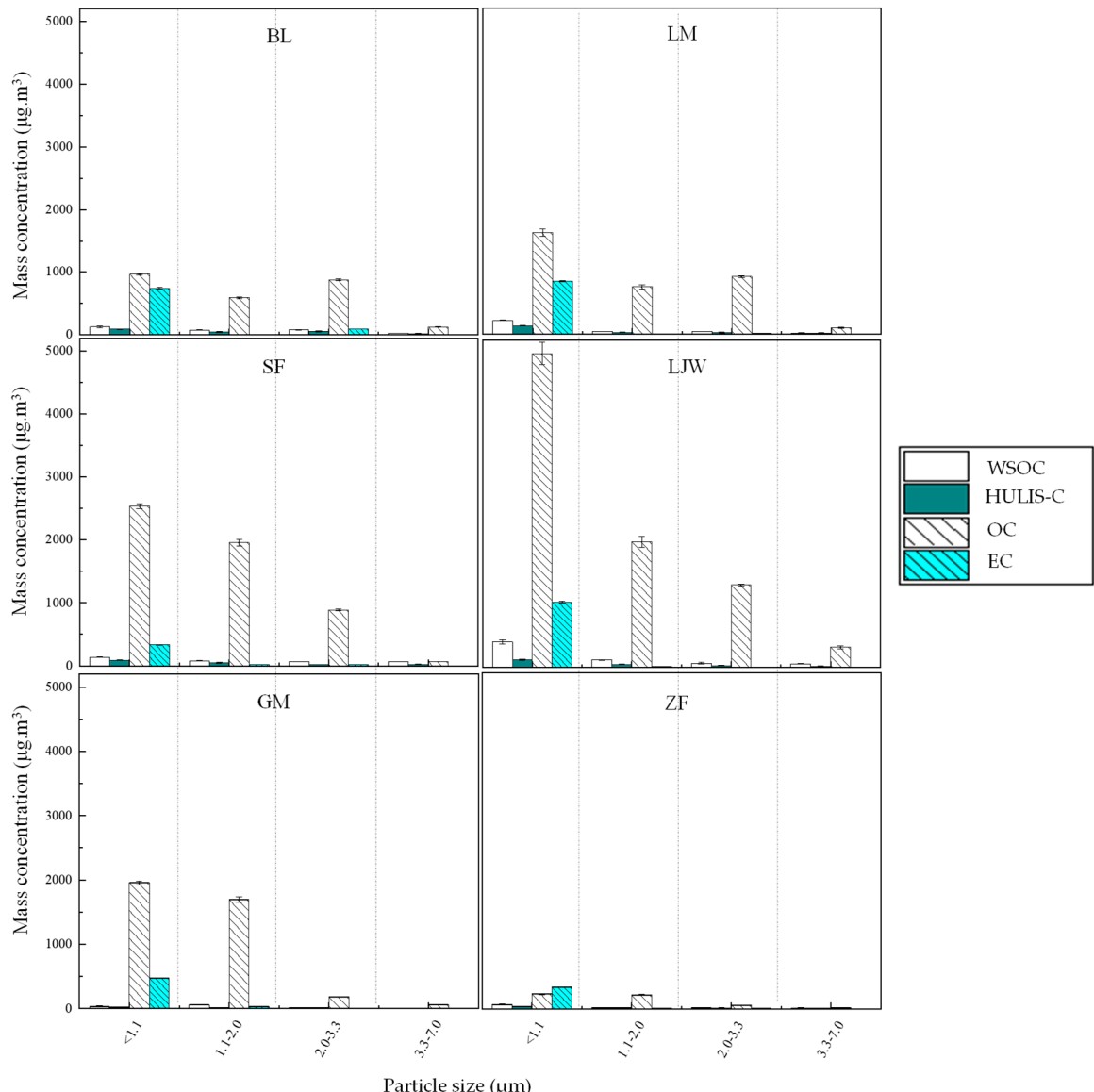

**Figure 7.** Mass concentration ($\mu g\ m^{-3}$) and standard deviation (STD) of OC, EC, WSOC, and HULIS-C (Coal, N = 6) in RCC particles. (Bole coal (BL), Luomu coal (LM), Shunfa coal (SF), Lijiawu coal (LJW), Guangming coal (GM), and Zongfan coal (ZF)).

**Table 6.** The percentage of OCx, ECx, WSOCx to WSOC, and HULIS-C in $PM_{2.0}$ and $PM_{2.0~7.0}$ from RCC particles.

| Samples | Size (µm) | HULIS-Cx/ WSOCx | OCx/ PMx | ECx/ PMx | WSOCx/ TCx | HULIS-Cx/ TCx | OCx/ ECx | HULIS-Cx/ PMx | WSOCx/ PMx | HULISCx/ HULIS-Ct |
|---|---|---|---|---|---|---|---|---|---|---|
| BL |  | 66.76 | 42.61 | 20.52 | 8.95 | 5.98 | 2.08 | 3.77 | 5.65 | 65.73 |
| LM |  | 61.11 | 30.37 | 10.84 | 8.52 | 5.21 | 2.80 | 2.15 | 3.51 | 79.30 |
| SF | $PM_{2.0}$ | 63.68 | 43.58 | 3.44 | 4.45 | 2.83 | 12.69 | 1.33 | 2.09 | 74.92 |
| LJW |  | 32.73 | 59.01 | 8.93 | 6.61 | 2.16 | 6.61 | 1.47 | 4.49 | 78.95 |
| GM |  | 43.07 | 87.18 | 12.02 | 2.17 | 0.93 | 7.25 | 0.93 | 2.15 | 68.48 |
| ZF |  | 55.73 | 24.41 | 19.03 | 11.30 | 6.30 | 1.28 | 2.73 | 4.91 | 76.34 |
| Average |  | 53.85 | 47.86 | 12.46 | 7.00 | 3.90 | 5.45 | 2.06 | 3.80 | 73.95 |
| TSD |  | 12.12 | 20.70 | 5.84 | 3.02 | 2.03 | 3.93 | 0.96 | 1.35 | 5.13 |
| BL |  | 66.20 | 40.12 | 4.36 | 9.79 | 6.48 | 9.20 | 2.88 | 4.35 | 34.27 |
| LM |  | 60.47 | 42.82 | 0.76 | 6.99 | 4.22 | 56.46 | 1.84 | 3.04 | 20.70 |
| SF | $PM_{2.0-7.0}$ | 33.91 | 41.44 | 0.98 | 13.79 | 4.68 | 42.11 | 1.98 | 5.85 | 25.08 |
| LJW |  | 35.17 | 46.76 | 0.48 | 8.00 | 2.81 | 98.36 | 1.33 | 3.78 | 21.05 |
| GM |  | 82.27 | 18.66 | 0.46 | 8.65 | 7.12 | 40.53 | 1.36 | 1.65 | 31.52 |
| ZF |  | 64.35 | 6.68 | 0.73 | 28.13 | 18.10 | 9.16 | 1.34 | 2.09 | 23.66 |
| Average |  | 57.06 | 32.75 | 1.30 | 12.56 | 7.23 | 42.64 | 1.79 | 3.46 | 26.05 |
| TSD |  | 17.32 | 14.75 | 1.38 | 7.29 | 5.06 | 30.39 | 0.55 | 1.41 | 5.13 |

(Bole coal (BL), Luomu coal (LM), Shunfa coal (SF), Lijiawu coal (LJW), Guangming coal (GM), and Zongfan coal (ZF)).

OC to EC ratio is commonly regarded as a secondary organic aerosol indicator when the OC/EC ratio exceeds 2.8 [60]. The OC/EC ratio decreases gradually with increasing particle size in the range of >1.1 µm. However, when the particle size <1.1 µm, the OC/EC ratio suddenly decreases to a minimum of 0.68. This phenomenon can be explained by the fact that due to insufficient combustion of coal in the initial stage of combustion [54], a large amount of black smoke is released, resulting in a lower ratio of OC/EC in <1.1 µm particles which is smaller than other particle sizes. The average contribution of WSOC and HULIS-C to the TC of RCC particles was 2.54–12.93% and 1.29–7.44% in $PM_{7.0}$.

### 3.3.3. Correlation of HULIS-C and WSOC with Other Species in RCC Particles

HULIS-C and WSOC can be produced from both primary emissions and secondary formations. Table 7 shows the correlation between HULIS-C and WSOC with water-soluble ion species. During our measurements, the concentrations of HULIS-C and WSOC were significantly correlated with SIA ($SO_4^{2-}$, $NO_3^-$, and $NH_4^+$), $K^+$, and $Mg^{2+}$ in RCC particles. The *p* values of Pearson correlation coefficients between HULIS-C and WSOC and water-soluble ions in RCC particles are shown in Table S4.

**Table 7.** The correlation between HULIS-C and WSOC and water-soluble ions in RCC particles.

| Samples | BL HULIS-C | BL WSOC | LM HULIS-C | LM WSOC | SF HULIS-C | SF WSOC | LJW HULIS-C | LJW WSOC | GM HULIS-C | GM WSOC | ZF HULIS-C | ZF WSOC |
|---|---|---|---|---|---|---|---|---|---|---|---|---|
| $Cl^-$ | 0.66 ** | 0.71 *** | 0.42 * | 0.44 * | 0.06 ** | 0.14 *** | 0.76 ** | 0.80 * | −0.74 ** | −0.77 ** | 0.13 ** | 0.19 ** |
| $NO_2^-$ | 0.06 ** | 0.14 *** | 0.15 * | 0.20 * | 0.94 ** | 0.99 *** | 0.63 ** | 0.72 * | −0.19 ** | −0.72 ** | −0.26 ** | −0.21 ** |
| $NO_3^-$ | 0.84 ** | 0.83 *** | 1.00 * | 0.99 * | 0.94 ** | 0.99 *** | 0.95 ** | 0.99 * | 0.58 ** | 0.07 ** | 0.94 ** | 0.98 ** |
| $SO_4^{2-}$ | 0.74 ** | 0.72 *** | 0.97 * | 0.97 * | 0.81 ** | 0.85 *** | 0.95 ** | 0.99 * | 0.72 ** | 0.49 ** | 0.98 ** | 0.99 ** |
| $Na^+$ | 0.89 ** | 0.80 *** |  | 0.99 * | 0.96 ** | 0.99 *** |  | −0.28 * | 0.73 ** | −0.29 ** | 0.97 ** | 0.99 * |
| $NH_4^+$ | 0.89 ** | 0.92 *** |  |  | 0.96 ** | 1.00 *** |  |  | 0.73 ** | 0.27 ** | 0.97 ** | 0.99 * |
| $K^+$ | 0.83 ** | 0.82 *** | 0.90 * | 0.89 * | 0.95 ** | 0.99 *** | 0.93 ** | 0.95 * | 0.69 ** | 0.30 ** | 0.97 ** | 0.99 ** |
| $Mg^{2+}$ | 0.78 ** | 0.76 *** | 1.00 * | 0.99 * | 0.93 ** | 0.98 *** | 0.93 ** | 0.96 * | 0.80 ** | 0.38 ** | 0.98 ** | 0.99 ** |
| $Ca^{2+}$ | 0.78 ** | 0.76 *** | −0.14 * | −0.09 * | −0.34 ** | −0.46 *** | −0.49 ** | −0.48 * | 0.29 ** | 0.90 ** | 0.98 ** | 0.99 ** |

***: means *p* value < 0.01; **: means *p* value 0.01–0.05; *: means *p* value > 0.05. (Bole coal (BL), Luomu coal (LM), Shunfa coal (SF), Lijiawu coal (LJW), Guangming coal (GM), and Zongfan coal (ZF)).

$NO_3^-$, $SO_4^{2-}$, and $NH_4^+$ are usually referred to as secondary water-soluble ions [9]. In this study, both HULIS-C and WSOC were strongly correlated with secondary water-soluble ions such as $NH_4^+$ (r = 0.89 ± 0.10), $NO_3^-$ (r = 0.88 ± 0.14, *p* < 0.05), and $SO_4^{2-}$ (r = 0.86 ± 0.11, *p* < 0.05) for HULIS-C and $NH_4^+$ (r = 0.80 ± 0.30, *p* < 0.05), $NO_3^-$ (r = 0.81 ± 0.33, *p* < 0.05), and $SO_4^{2-}$ (r = 0.83 ± 0.18, *p* < 0.05) for WSOC, as shown in Table 7. $Cl^-$, $NO_2^-$, $Na^+$, and $Ca^{2+}$ exhibited a relatively weak link with HULIS-C and WSOC. Table S5 shows that during simulated RCC with high relative humidity (79 ± 2.09%), heterogeneous reactions of $SO_2$ and NOx happened more easily [61]. This finding may indicate that HULIS-C and WSOC are subject to a similar formation process

as SIA through secondary formation processes [61,62]. Relevant studies have shown that HULIS may result from primary emission sources and secondary formations involving newly formed sulfate particles [7,63].

## 4. Human Health Risk of WSPTMs via Inhalation Exposure

HULIS chelate transition metals in atmospheric particulate matter and participates in the redox cycle [13], such as Copper (Cu) and Iron (Fe), may cause oxidation potential in lung fluid [64]. In the HULIS-Fe (II) system, the ROS generation capacity depends on the mixing time of HULIS-C with Fe (II), and the Fe (II)-induced Fenton reaction plays a role in cell mortality [13]. Cu, Ni, Fe, Cr (VI), Co, As, Mn, V, and Zn could support electron exchange [65] and induce the formation of reactive oxygen species in the lungs [66], causing damage of oxidative DNA and inflammation of respiratory tracts [67].

The *HQ* of V, Cr (VI), Co, As, Cd, Zn, Mn, and Ba within $PM_{1.1}$, $PM_{1.1 \sim 2.0}$, $PM_{2.0 \sim 3.3}$, $PM_{3.3 \sim 7.0}$, and $PM_{>7.0}$ in RCC particles were estimated (Table S6). Among these PTMs, Ba, As, and Mn had the greatest contributions to *HI*, with risk (contributions) of $3.66 \times 10^0 \pm 3.28 \times 10^0$ ($78.39 \pm 19.53\%$), $2.34 \times 10^{-2} \pm 8.61 \times 10^{-2}$ ($11.89 \pm 10.20\%$), and $6.38 \times 10^{-2} \pm 1.84 \times 10^{-2}$ ($2.57 \pm 1.39\%$) in TSP, respectively. The *HI* of PTMs in descending order was Ba > As > Mn > Ni > Co > Cd > V > Cr (VI) in $PM_{1.1}$. The *HQ* values of WSPTMs were all below one within $PM_{1.1 \sim 2.0}$, $PM_{2.0 \sim 3.3}$, $PM_{3.3 \sim 7.0}$, and $PM_{>7.0}$ and were higher than one in $PM_{1.1}$, indicating that there was insignificant non-carcinogenic risks for both children and adults in $PM_{1.1}$ and zero non-carcinogenic risks for both children and adults within $PM_{1.1 \sim 2.0}$, $PM_{2.0 \sim 3.3}$, $PM_{3.3 \sim 7.0}$, and $PM_{>7.0}$ via inhalation exposure for each WSPTMs of RCC particles [68].

The *CR* of V, Cr (VI), Co, As, Cd, and Pb for children and adults within $PM_{1.1}$, $PM_{1.1 \sim 2.0}$, $PM_{2.0 \sim 3.3}$, $PM_{3.3 \sim 7.0}$, and $PM_{>7.0}$ in RCC particles were estimated (Table S7). As expected, most of the toxic metals exhibited high *CR* values in the smaller particles (<1.1 μm) because of their high deposition efficiencies [30]. The *TCR* of PTMs for adults and children in the smaller particles (<1.1 μm) contributed $42.95 \pm 8.49\%$ and $42.01 \pm 8.72\%$ to the TSP, respectively. Among both children and adults, the *CR* of As was the highest ($1.40 \times 10^{-6} \pm 5.16 \times 10^{-7}$ for children, $5.60 \times 10^{-6} \pm 2.06 \times 10^{-6}$ for adults) followed by Cr (VI) ($1.40 \times 10^{-6} \pm 3.53 \times 10^{-6}$ for children, $5.60 \times 10^{-6} \pm 1.41 \times 10^{-6}$ for adults) in TSP. Pb had the lowest *CR*, which was approximately $1.47 \times 10^{-2}$ times for children and $7.21 \times 10^{-1}$ for adults of As in TSP. The *TCR* values (which reached $4.04 \times 10^{-6} \pm 8.18 \times 10^{-7}$ for children and $1.52 \times 10^{-5} \pm 2.99 \times 10^{-6}$ for adults) exceeded the acceptable level ($1 \times 10^{-6}$), indicating that we should pay more attention to these WSPTMs.

## 5. Conclusions and Perspectives

### 5.1. Conclusions

Lung cancer has unique epidemiological characteristics due to the toxicity of indoor RCC particles in Xuanwei [69–71], which suggests that there may be unique molecular mechanisms for the development of lung cancer in Xuanwei. However, the mechanism of the high lung incidence is still not clear. In this study, we selected six types of coal and conducted simulated combustion experiments to explore the content and particle size distribution pattern of HULIS-C in particulate matter produced by RCC, providing new perspectives and evidence to reveal the high incidence of lung cancer in Xuanwei.

(1) The particulate matters emitted from RCC were mainly distributed in the size range of <2.0 μm, which accounted for 52.45–77.16% of total particulate matter, which is higher than other areas of China.

(2) HULIS-C to the PM was 2.09–5.65% for $PM_{2.0}$ and 2.68–5.62% for $PM_{2.0–7.0}$, respectively. HULIS-C emitted from RCC was mainly concentrated in $PM_{2.0}$ (68.48–79.30%)

(3) During our measurements, the concentrations of HULIS-C and WSOC were significantly correlated with $SO_4^{2-}$, $NO_3^-$, and $NH_4^+$ in RCC particles.

(4) HULIS-Cx to HULIS-Ct (%) values in RCC particles were 68.48–79.30% (average $73.95 \pm 5.13\%$) for $PM_{2.0}$ and 20.70–34.27 (average $26.05 \pm 5.13\%$) for $PM_{2.0–7.0}$, respectively. The HULIS-Cx to WSOCx (%) values in RCC particles were 32.73–63.76%

(average 53.85 $\pm$ 12.12%) for PM$_{2.0}$ and 33.91–82.67% (average 57.06 $\pm$ 17.32%) for PM$_{2.0-7.0}$, respectively.

(5)  *TCR* values exceeded the acceptable level ($1 \times 10^{-6}$), indicating that we should pay more attention to these PTMs. For both children and adults, the CR of As was the highest followed by Cr (VI), and Pb had the lowest *CR*.

### 5.2. Perspectives

In the future, a comprehensive investigation of coal combustion HULIS-C emissions under different stove types, combustion conditions, and combustion stages is necessary to better understand HULIS-C. Unfortunately, HULIS-C is a powerful sequestering agent in atmospheric particulate matter and there is a lack of information on the ROS generated [4] by the HULIS-metal combination through the cellular matrices and tissue. Some attempts should be done in cell-free and cell-based experiments to obtain well-characterized information about the ROS generated by the HULIS-metal combination and to better address the health effects of HULIS-C.

### 5.3. Policy Implications

The following policy suggestions can be drawn based on the findings in our study.

(a)  The government should increase awareness of environmental protection in rural areas.
(b)  The government should improve the structure of houses and fireplaces in the area.
(c)  In order to effectively mitigate the severe PM$_{2.5}$ pollution and promote environmental equity, a differentiated carbon policy should be considered. The local government should better adjust the structure of domestic energy, reduce the use of coal and biomass, and promote the use of environmentally friendly sources in Xuanwei area.

**Supplementary Materials:** The following are available online at https://www.mdpi.com/article/10.3390/pr9071254/s1, Figure S1: The correlation coefficients between NH$_4^+$ and SO$_4^{2-}$, Table S1: The mass concentration of ion species ($\mu$g m$^{-3}$) and percentage of SIA/WSIIs (%), Table S2: Charge balance of ions, Table S3: Pearson correlation coefficients between HULIS-C and WSOC, Table S4: The *p* value of Pearson correlation coefficients between HULIS-C and WSOC and water-soluble ions in RCC particles, Table S5: Simulated combustion conditions, Table S6: Non-carcinogenic health risk of WSPTMs in RCC particles, Table S7; Carcinogenic health risk of WSPTMs in RCC particles.

**Author Contributions:** Conceptualization, K.X., S.L. and Q.W.; methodology, S.L., W.W., Y.L. and S.Y.; software, K.X.; validation, W.W. and S.L.; formal analysis, K.X. and Q.W.; investigation, Q.W. and S.L.; resources contribution, Q.W.; data curation, K.X. and Q.W.; writing—original draft preparation, K.X.; writing—review and editing, K.X., S.L. and Q.W.; visualization, K.X.; supervision, K.X. and Q.W.; project administration, Q.W.; funding acquisition, Q.W. All authors have read and agreed to the published version of the manuscript.

**Funding:** This study was partially supported by the Special Funds for Innovative Area Research (No. 20120015, FY 2008-FY2012) and Basic Research (B) (No. 24310005, FY2012-FY2014; No. 18H03384, FY2017~FY2020) of Grant-in-Aid for Scientific Research of Japanese Ministry of Education, Culture, Sports, Science and Technology (MEXT), and the Steel Foundation for Environmental Protection Technology of Japan (No. C-33, FY 2015- FY 2017).

**Institutional Review Board Statement:** Not applicable.

**Informed Consent Statement:** Not applicable.

**Acknowledgments:** We would like to express gratitude to Shanghai University of Electric Power, where we tested the WSOC and HULIS-C.

**Conflicts of Interest:** The authors declare that they have no conflict of interest.

## Abbreviations

| | |
|---|---|
| AT | Average lifetime |
| $NH_4^+$ | Ammonium ion |
| BL | Bole |
| CR | Carcinogenic risks |
| $Ca^{2+}$ | Calcium ion |
| $Cl^-$ | Chlorine ion |
| CMAQ | Community Multiscale Air Quality |
| EC | Elemental carbon |
| US EPA | Environmental Protection Agency |
| ED | Exposure duration |
| EF | Exposure frequency |
| ET | Exposure time |
| HQ | Hazard quotient |
| HULIS | Humic-like substances |
| HULIS-C | HULIS containing carbon contents |
| GM | Guangming coal |
| LJW | Lijiawu coal |
| LM | Luomu |
| $Mg^{2+}$ | Magnesium ion |
| $NO_3^-$ | Nitrate ion |
| $NO_2^-$ | Nitrite ion |
| IUR | Inhalation unit risk |
| RfC | Inhalation reference dose |
| OM | Organic Matter (OM) |
| $K^+$ | Potassium ion |
| RCC | Residential coal combustion |
| ROS | Reactive oxygen species |
| HI | Total hazard quotient |
| TCR | Total carcinogenic risks |
| SIA | Secondary inorganic aerosol |
| SF | Shunfa coal |
| $Na^+$ | Sodium ion |
| $SO_4^{2-}$ | Sulfate ion |
| WSOC | Water-soluble organic carbon |
| WSPTMs | Water-soluble potentially toxic metals |
| WSIIs | Water-soluble inorganic ions |
| ZF | Zongfan coal |

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
