# Peer review of "Approval Research for Carcinogen Humic-Like Substances (HULIS) Emitted from Residential Coal Combustion in High Lung Cancer Incidence Areas of China"

_processes, doi:10.3390/pr9071254_

Round 1

Reviewer 1 Report

Dear authors,

I suggest extending the literature review section and highlight methodological differences between other connected papers and yours. Please consider including the significance levels corresponding to the results from Table 7. I suggest spotlighting the main implications for local decision makers: how should they make best use of the results of your research.

Reviewer 2 Report

The article “Approval research for carcinogen humic-like substances (HULIS) emitted from residential coal combustion in high lung cancer incidence areas of China” explores the linkage between lung cancer mortality rate in Yunnan Province of China was due to the emissions coming from residential coal combustion (RCC).

Overall the article brought up a case study that is of great importance. I have a number of suggestions/recommendations which could be considered before the final acceptance of the article:

-          The introduction section needs a better connection between the sentences and paragraphs.

-          Some of the references in the introduction section (i.e. lines 38 & 41) are not correctly written, see the journal guideline.

-          A paragraph on the rationale of the study is missing in the introduction section.

-          There are typing/grammatical errors in the paper, see Table 3 (Don’t know what means, Vale of adults).

-          Section 2.6: No description required for the common tools, i.e. MS Excel. 

-          Some of the terminologies used are i.e. concentrations of ion species are not clearly described (see line 234).

-          Line 238, Secondary inorganic aerosol (SIA), Sulphate (SO4-2) ……….

-          The quality of figure 6 is poor, authors should replace the figure with a clear x-axis.

-          Table 7. The statistically correlated values should be presented in bold.

I would like to accept this article after the suggested changes. 

Reviewer 3 Report

The article raises a very important topic about human life. However, it lacks a decent literature introduction, which is necessary to justify the research conducted. Especially when it comes to HULIS, which is needed to understand the need for this article.
I also suggest a more extensive description of the conclusions.

Round 2

Reviewer 1 Report

The authors have improved the quality of the manuscript.